# Oncological Diseases and Social Costs Considerations on Undertaken Health Policy Interventions

**DOI:** 10.3390/ijerph17082837

**Published:** 2020-04-20

**Authors:** Tomasz Holecki, Maria Węgrzyn, Aldona Frączkiewicz-Wronka, Karolina Sobczyk

**Affiliations:** 1Department of Health Economics and Management, Faculty of Health Sciences in Bytom, Medical University of Silesia in Katowice, 41-902 Bytom, Poland; kesobczyk@gmail.com; 2Department of Finances, Wroclaw University of Economics and Business, 53-345 Wrocław, Poland; maria.wegrzyn@ue.wroc.pl; 3Department of Public Management, University of Economic in Katowice, 40-287 Katowice, Poland; afw@ue.katowice.pl

**Keywords:** social costs in healthcare, indirect costs in healthcare, maps of health needs, access to oncological services, financing of medical infrastructure, health policy

## Abstract

The growing incidence and prevalence of civilization diseases is prompting national and transnational entities to seek instruments that would reverse epidemiological trends. Not without significance is the need to design such solutions that are going to provide an improved relation between the costs incurred to maintain health or recovery and the profit for citizens of continuing to function in good health. In its strategic documents, the European Union indicates the most important development goals in each financial perspective and the tools necessary to achieve them. In the Europe 2020 strategy, a cohesion policy was indicated as an important tool for the implementation of development goals, focusing on supporting activities leading to the equalisation of economic and social conditions in all regions of EU countries. The implementation of one of the three basic priorities of the Europe 2020 strategy, which is inclusive growth—supporting an economy with a high level of employment and ensuring social and territorial cohesion—assumes, among others, that in 2020, the population at risk of poverty and social exclusion will decrease by 20 million and that the employment rate in the EU will increase to 75%. Meeting the objectives will not be possible without a holistic coordinated approach to healthcare at the national and regional level in accordance with the principle of “health in all policies”. It also requires the involvement of various sources of financing, including structural funds. The EU’s prioritisation of the problems related to ensuring decent conditions for achieving health resulted in the mobilisation of structural funds for actions taken in the healthcare sector. Of particular importance are those actions which are taken to prevent, alleviate, and prevent oncological diseases. An additional contribution to undertaking actions aimed at preventing oncological diseases are the high and often neglected social costs incurred by societies. The goal of the article was to identify and evaluate actions taken in this area in Poland. It was achieved by analysing the literature on the subject and statistical data, and conducting induction based on the above-mentioned sources.

## 1. Introduction

Over the centuries, global morbidity and mortality have gradually changed the direction and thinking of policy makers shaping healthcare policies. They also contributed to initiating steps to stop the growing morbidity and reduce mortality as a consequence of the search and application of modern solutions in the field of mitigation and prevention of disease. Protective and preventive actions have contributed to the increasing number of people who have escaped death due to illness in the global population, although their current state of health and the need for constant monitoring has become a major challenge for healthcare systems. Due to the importance of health as both an individual and social value, individual countries assume the responsibility to care for the health of citizens, especially since health and education are the key elements of creating human capital, which is the most important factor in the economic growth of modern economies [1]. Therefore, one cannot underestimate the problem of creating systemic safeguards for health, but instead must constantly strive to introduce solutions aimed at maintaining human potential at the highest level. This is confirmed by the results of the research carried out by the World Bank, which showed the strength of health’s impact on the income growth rate and its size [2]. For example, a five-year extension of life expectancy leads to an increase in the annual GDP growth rate of 0.3%–0.5% [3]. A significant economic development brake is, however, unnecessarily incurred as too high indirect costs calculated as uncreated GDP caused by disease or ineffective treatment [4]. This group of costs should be well identified and analysed in order to minimise it or eliminate it completely. Particularly high costs of this nature are incurred in the context of new diseases for which the world has not yet discovered effective treatments. Premature deaths and a limited possibility of performing work due to health damage directly affect the value of these costs. Oncological diseases constitute one such difficult group of diseases.

In numerous countries, oncology is becoming a priority of the implemented health policy, and in view of the staggering costs of diagnosing and treating cancer, the need to increase the efficiency of the use of available resources becomes evident. In 2018, the standardised total cancer incidence rate reached a global value of 197.9/100.000. A higher incidence was recorded for men (218.6/100,000) than women (182.6/100,000). Equally important for assessing the incidence are studies related to individual regions of the world [5,6].

The highest overall cancer incidence rates for both sexes were recorded in 2018 in Australia (468/100,000), New Zealand (438.1/100,000), and Ireland (373.7/100,000). In the case of male incidence, these are the same three countries with values: 579.9/100.000 (Australia), 526/100.000 (New Zealand), and 430.8/100.000 (Ireland). The highest rates of cancer incidence among women were recorded in Australia (363/100,000), New Zealand (358.3), and Hungary (330.6/100,000) [7]. In 2018, there were about 18 million cancer cases worldwide, of which about 9.5 million were male and 8.5 million were female. Lung and breast cancers were the most commonly diagnosed oncological diseases, each of which constituted 12.3% of the total number of new cases diagnosed during the analysed period. Colorectal cancer was the third most common cancer (10.6% of all diagnoses). For men, the most commonly diagnosed cancers were lung cancer (15.5%), prostate cancer (14.5%), and colon cancer (11.4%). In turn, for women, these were breast cancer (24.5%), colon cancer (9.7%), and lung cancer (8.8%) [7].

According to estimates, the number of new cases of cancer will increase annually in the world over the next 20 years by over 63% from about 18 million in 2018 to over 29.5 million in 2040. For men, this will be an increase of almost 70%; in the case of women, almost 57%. It is estimated that the number of deaths in the analysed period will increase by over 71% in both sexes from approximately 9.5 million in 2018 to over 16 million in 2040. Mortality for men will increase by approximately 75%, for women, by approximately 68% [8]. The analysis of changes in the burden of disease in Poland in 1990–2017, compared to other Central European countries, is part of the previously presented trends [9].

These dynamic trends of morbidity and deaths directly point to the emergence of huge losses both in terms of population size and lost productivity understood as a limitation of the possibility of performing work due to health damage. This last statement means that it is necessary to strive to limit these losses by increasing clinical effectiveness in the ongoing treatment process but also by minimising non-medical costs arising during the disease. Increasingly perfect methods of estimating costs incurred by sick people are used to limit the losses.

## 2. The Problem of Social Cost Definition and Identification 

The analysis of the cost calculation process should not be limited to taking into account the costs of using specific health technologies, i.e., the amount of expenditure related to treatment and possibly prevention. All areas through which the disease may affect individual economic entities should be included in such an analysis. In fact, the effects of the disease are felt by all participants in economic life, affecting households and enterprises, and their effects can be experienced at the level of the entire economy. Therefore, cost analysis beyond the cost of benefits is becoming more and more popular among scientists but also among the governments of countries responsible for maintaining health systems. Further decisions taken by countries on the directions of the development of healthcare systems depend on the results of these analyses. Therefore, the question arises about the method of recognition and the definitions of individual types of costs accepted for these calculations. The variety of approaches adopted by researchers in the literature is quite wide. Table 1 presents the classification of costs, illustrating various approaches.

Thus, economic cost is understood differently in the perspective of the economic entity, as a microeconomic cost, and in the social perspective. In the social perspective, the economic cost is equivalent to the social cost. However, the social cost itself is also interpreted differently in the literature. One can distinguish the general approach, the broad approach, the narrow approach, and the WHO approach. The social cost component known as indirect costs occurs in the broad approach (as lost productivity) and in the WHO approach (as lost production). It also means that indirect costs should be interpreted as a category of lost social benefits (in the interpretation of externalities) or otherwise an alternative social cost (expressed as lost productivity, lost production).

It is also worth noting the fact that friction costs, i.e., lost benefits (opportunity costs), estimated from the entrepreneur’s perspective, are not the same as opportunity costs in the economic costs account in the management account kept in the enterprise. The management account does not count lost profit due to absenteeism or inability to work but only due to not using another alternative to make a profit.

In the economy, there are three approaches to estimating indirect costs (understood as a component of social cost in a broad sense), depending on the adopted theoretical concept: From the human capital approach, the friction cost method and taking into account the assessment of health as the willingness-to-pay-method [29]. Each method shows the cost of the time unit and the length of the period of reduced productivity. The leading parameter is therefore the period of reduced productivity, which is particularly reflected in chronic diseases. The implementation of measures to support chronically ill people in leading an active and productive life has great social and economic value [29].

The human capital method, as the most commonly used, in practice is firmly rooted in economic theory. It includes a cost-based and revenue-based approach. The precursor of the cost-based approach is F. Engel (1883) [30]. In this method, one takes into account all expenses related to the subsistence and upbringing of a person, treating them as manufacturing costs. Despite its popularity, this method is often criticised due to attributing different productivity to individuals depending on age and gender, and thus being discriminatory, disregarding business cycle phases or simply overestimating the costs resulting from a permanent inability to work or death. It is worth noting that these reservations do not refer, however, to the method itself but to the imperfections of the employed measures of lost productivity. Supporting the method by considering the coefficients of professional activity or life expectancy tables in the calculations will successfully reduce the identified shortcomings. This method is still being improved and has been developed in many scientific works, e.g., by L. I. Dublin and A. J. Lotka (1997) [31].

The second of these methods related to the estimation of indirect costs is the method of friction costs. It presents the entrepreneur’s perspective, where it is assumed that the labour resources in the economy are not being fully used. The indirect cost is not the total loss of capital value that is lost due to the patient’s inability to work but only the part that will not be worked out by other people or by the patient after returning to full health. This assumption is based on a partial failure to perform the assumed obligations in connection with the occurrence of a short absence from work. Only an urgent part of the patient’s duties is usually transferred to other employees, in whole or in part, and the rest awaits his return to work. Adopting the entrepreneur’s perspective means that in the method of friction costs in the event of long-term illness or death of the sick, indirect costs are incurred only for the time necessary until the enterprise returns to the initial level of production (the so-called friction period) [29,32].

The last of the significant methods for estimating indirect costs is the willingness to pay method. It is mainly used for the valuation of goods and services that do not have their market price. Loss of production is expressed by the monetary value that a person would be able to pay for reducing the likelihood of illness or premature death. Estimating the value in this method is based on revealed preferences, i.e., estimating the value of human actions taken to reduce the risk of an adverse event or declared preferences, considering the responses of respondents about previous choices. This method is mainly based on surveys, but due to difficulties in identifying and subjectively assessing indirect costs (e.g., pain assessment or loss of free time), it is often considered to be unreliable.

The outlined methods for estimating the costs of lost production require the adoption of a unit of measure of labour productivity. Due to the multitude of approaches and difficulties with its homogeneous measurement, we found many measures of productivity in the literature. Among them, the simplest is the society’s wealth indicator, expressed by the gross domestic product (GDP), a de facto measure of productivity, and its multiple variations. If both the quantity and quality of services rendered are determined by the national potential, then the size of this aggregate and its portion allocated for healthcare affects the quality of life of patients and potential patients who, aware of the level of medical care and its availability, feel either calm or anxious. When entering the normative economy of welfare and the dispute about the fair distribution of goods, it is known that a homogeneous indicator cannot be presented, but it can be estimated and subjectively estimated. The solution may be a construction modelled on the index of the economic aspects of welfare, which tries to deduce the level of wealth of the studied social group from the set of information about the consumption of goods, by probing, among others: The level and proportion of expenditure on consumption, investment, leisure time value, value work in the household, expenditure on healthcare (in various calculation periods), educational expenditure, etc. Meters used, e.g., by the United Nations, are also available for use, including HDI (human development index) and HPI (human poverty index) inferring the population’s situation based on GDP per capita, forecasting life expectancy at birth, or analysing the level of citizens’ education. Additional information that seems important from the point of view of the issue of estimating the costs of a disease entity is, among others: Relatively easily measurable income from work or other financial benefits, financial state transfers, non-public transfers, informal payments in healthcare, degree of resignation from processes of treatment for financial reasons, resignation from buying coordinated medicines, degree of use of non-public healthcare, refusal to provide services (limits, queues, etc.), and individual funding of hospital stay (additional tests, medicines, meals, etc.).

Because the level of affluence of the population directly affects the ability to absorb medical services and the quality of life of the diagnosed population, the analyses can also be based on information on household income and data of a similar nature. In practice, the most frequently used measures of productivity in both economics and pharmacoeconomics include gross remuneration and gross domestic product per capita. Each of these measures allows expression of the volume of production in monetary units and is widely accepted and understood. In view of the above, information on the amount of expenditure on health care, with a particular emphasis on the selected prevalence area, is very much needed and indicates the necessary directions for the future. The level of expenditure, considering one of the above measures, for selected highly developed countries is presented in Table 2.

There are numerous comparisons in the literature regarding the area of healthcare expenditure implemented in various countries. They differ in the arrangement of groups of countries or the indicators adopted for research [33]. Nevertheless, they all show a large discrepancy in the amount of funds allocated to healthcare.

However, as shown in Table 2, the differences in the value of financial resources transferred by governments to oncological care are very large. This is not due to a significant difference in incidence in individual countries, but to the systemic solutions adopted and national financial capabilities. Raising the role of prevention, health promotion and disease management can significantly reduce the adverse effects of chronic diseases on the environment in which the patient resides, as well as on the entire economy. The social costs of diseases will not be so burdensome and will certainly decrease. Consequently, the problem of productivity lost due to illness will most likely have less and less significance.

For this reason, numerous researchers have published papers presenting analyses related to the components of social costs understood mainly as indirect costs covering lost productivity. However, the methodology used to conduct these studies is usually very different. The issue of indirect costs (as defined above) is usually only an element to the considerations that mainly show the analysis of financial recommendations used in the treatment of specific diseases. For example, it is possible to calculate works in the field of assessment of multiple sclerosis for Poland, covering the values of direct costs (medical and non-medical) and indirect costs (as lost productivity) [34]; in the field of Crohn’s disease for Poland, including the values of direct costs (medical and non-medical) and indirect costs (as lost productivity) [35]; in the field of colorectal cancer for Serbia, including direct costs (medical and non-medical) and indirect costs (as lost productivity) [36]; in the field of breast cancer, colorectal cancer, and non-Hodgkin’s lymphoma for the countries of south-eastern Europe, covering essentially only the direct costs (medical and non-medical) [37]; in the scope of selected 29 disease groups from oncological treatment for Serbia, also including only the direct costs [38]; or costs of oncological diseases related to direct costs (radiology performance) for Serbia [39].

At the same time, interesting comparative research has been conducted for selected countries from the OECD group, covering indirect costs understood as costs of absence from work and costs of benefits, and related to incidence in general. Against this background, oncological diseases stand out clearly, which is emphasized by the authors of the report, showing that the incidence of cancer can become the highest-rated factor causing disability and thus loss of productivity [40].

Therefore, the complexity of the goal of striving for clinical effectiveness and economic efficiency in cancer treatment is a very important challenge for health policy makers making decisions not only in healthcare. Current costs of oncological treatment in Poland amount to almost 10% of the budget of the Polish payer (for the National Health Fund) annually. In the group of most common cancers (analysis based on the list of diseases according to the international ICD-10 classification, namely: C18 colorectal cancer, C22 malignant liver and intrahepatic bile duct cancer, C34 malignant bronchial and lung cancer, C50 malignant tumour, C67 malignant tumour, C67 malignant bladder cancer, Hodgkin C81 disease, multiple myeloma, and C90 plasma cell tumours) in 2015 for the Lower Silesian Voivodeship, the direct medical cost incurred by the payer (Social Insurance Institution) was 242.9 million PLN, and in 2016, it was already 262 million PLN (an increase of 7.2%) [41]. By supplementing these values with the costs of preventive programs, estimated indirect costs, including the costs of lost production in the economy resulting from the occurrence of cancer, and the costs associated with the broadly understood social security, e.g., costs of sick leave, we can see that the level of total expenditure is many times higher. The expenses of the Polish payer of social security benefits (Social Insurance Institution) on social security benefits related to the above-mentioned most common cancers increased from 47.5 million PLN in 2014 to 53.5 million PLN in 2016 (up by as much as 11.2%). Indirect costs, understood as lost productivity (economic losses), amounted to 878.1 million PLN (i.e., 0.579% of GDP in 2015) and increased to 917 million PLN (i.e., 0.589% of GDP in 2016) [4]. Based on the forecasts of the number of cases and deaths due to malignant neoplasms published by the Polish Oncological Society, it was estimated that in 2025, the indirect costs of neoplastic diseases will be higher by approximately 29% compared to 2013 and will amount to approximately 1.3% of GDP [42].

## 3. Actions of Healthcare Politicians Focused on Limiting Social Costs for Cancer in Poland

Considering the increasing (economic and social) costs of oncological diseases and the consequences of these diseases, governments of many countries go to great lengths to introduce changes in their respective healthcare systems. There is a lot of international experience regarding the creation and implementation of cancer plans and strategies. National strategies or plans for combating cancer are of a different nature and form a collection of various activities that take into account their own needs and problems. Individual international organisations propose their own methodological approaches to creating strategies. The report of Imperial College in London shows that in 2009, out of the 27 European countries covered by the analysis, only 16 had strategies or national plans to combat cancer [41].

Health policy in Poland in the field of oncology is based on the Cancer Strategy 2015–2024 [42] (before 2015 on the National Strategy for Combating Cancer) and is one of the most important manifestations of the activity of state institutions undertaken to guarantee the highest possible range of health safety for citizens. Among the activities of public entities, the effects of which are related to the health of the population, one can indicate, among others, identifying health priorities for the country, or developing health needs maps. The relevant legal acts [43] list the following as priorities, among others: Reduction of incidence and premature mortality due to cardiovascular diseases (including heart attacks, heart failure, and strokes), malignant neoplasms, chronic respiratory diseases, and diabetes [44]. The selection of these priorities results from the assessment of the scale of the most frequent causes of diseases and premature deaths due to civilisation diseases constituting the greatest burden on the healthcare system and social security. Tumours are the second most common cause of death in Poland. More than 23% of cancer deaths are due to malignant tracheal, bronchial, and lung cancer; for another 17.3%, it is cancer of the stomach, colon, rectum, intestinal, and rectal junctions. Deaths due to breast cancer represent over 8% of all deaths due to cancer [45].

The priorities set in this way translated into the intensification of activities aimed at improving access to medical services. Consequently, for example, a special path has been defined for persons with oncological diagnosis, i.e., fast oncological therapy, the implementation of which has been allocated part of the financial resources from the general pool of health insurance contributions. The card entitling the holder to rapid diagnostics and oncological treatment (the so-called DiLO green card) was introduced in Poland on January 1 2015 [46] together with the oncological package, a group of provisions aimed at improving diagnostics and improving cancer treatment in Poland. Rapid oncological therapy is dedicated to patients in whom doctors suspect or find a malignant tumour and patients undergoing oncological treatment. There are no age restrictions on access to treatment as part of rapid oncological therapy. Diagnosis and treatment as part of the oncological package are also not covered by the limits set by the payer (National Health Fund), which is the rule for other services.

Within the framework of rapid oncological therapy, maximum deadlines were set for individual stages of treatment, which is an absolute novelty in the Polish healthcare system. The inspiration for these entries were solutions adopted and working in other countries. In many countries, there are guidelines and regulations defining the waiting time of patients for oncological treatment [47]. In the United Kingdom, this time is guaranteed by law: The maximum waiting time for a patient to see a specialist doctor (after visiting a General Practitioner (GP)/family doctor) is 14 days; and from the diagnosis of cancer to the beginning of treatment: 31 days. The maximum time from referral by a GP/family physician suspecting cancer to treatment is 62 days. Similarly, in Denmark, where the first visit to an oncologist specialist must take place within two weeks of receipt of the referral (referrals to the specialist are delivered electronically, the patient is also invited for a visit electronically or by telephone), and oncological treatment must start no later than in the next two weeks. The solutions adopted in Poland are similar. The statutory maximum initial diagnostics time is 35 days (28 days from 2016), and the maximum in-depth diagnostics time is 28 days. The time from conducting a consultation to starting treatment cannot exceed 14 days [48]. On the other hand, this reform does not specify the maximum time that can elapse from referral by a GP or another specialist who suspects the occurrence of cancer to the start of treatment.

Actions taken as part of rapid oncological therapy are intended to reduce the time of access to necessary treatment procedures. The task and expected results formulated in such a way are a sign of care for improving access to services that are part of the whole health policy pursued by the state. Unfortunately, however, in practice, the process started in 2015 has not yet affected the clear decline in access time to services. Currently, the average waiting time for oncology services in Poland is 2.2 weeks for a patient with a DiLO card and 6.6 weeks for a patient without a DiLO card [49]. The average waiting time for oncological services increased in four out of five monitored fields: Oncology, oncological surgery, haematology and haematooncology, and oncological radiotherapy. However, the average waiting time in oncological gynaecology decreased. This is of course due to the complicated diagnostic and therapeutic procedure in oncological procedures. Conducting a full process usually consists of a number of procedures, ranging from visiting a primary care physician, visiting a specialist, conducting in-depth diagnostics, convening a consultation, and implementing hospital treatment, and problems arising at individual stages affect the length of the treatment initiation process. In accordance with the assumption of the Ministry of Health, the process of diagnosing an oncological patient should not exceed seven weeks (from 2017). Meanwhile, as the latest WHC report [50] shows, the diagnostic process taking into account the stage of treatment exceeds seven weeks in the case of prostate cancer (11.9 weeks) and breast cancer (8.8 weeks).

The creation of maps of health needs as an important tool supporting decision-making processes is another manifestation of activity in the field of health policy. These maps are assumed to be an advanced analytical tool supporting management decisions in health care based on demographic and epidemiological trends, as well as existing infrastructure and forecast needs in this respect. The obligation to develop maps of the demand for medical services in the planning process and the distribution of funds for healthcare is assigned to the Ministry of Health. In the process of planning health needs maps, it was recognised that the assessment of the actual demand for health care services from specific types and ranges would lead to a better adjustment of financing health services in relation to the epidemiological and demographic situation in the country, as well as individual local government units. The maps include analyses of the health needs of individual regions and the whole country, demographic and epidemiological data, data on services rendered, as well as data on human resources and equipment. Such maps of health needs are also created in other countries, e.g., in Great Britain, France, or Denmark [51], and therefore, due to the proven effectiveness of this tool, the development of maps of health needs in Poland should be considered as acting in the right direction and employing good practices.

Pioneers in using the technique of mapping health needs to optimise decisions on the directions of allocation of public funds to meet health needs are countries, such as Sweden (identification of regional differences as a consequence of statistical analysis of defined indicators used to estimate health needs; in-depth analysis of selected health issues) [52,53,54], Austria (identification of the level of bed occupancy and length of stay of patients in hospital; qualitative and quantitative assessment of physical and personal resources in relation to the number and structure of hospitalisation; planning of hospital services as a consequence of patient flow analysis and resource restrictions) [55,56], Germany (identification of physical and personal resources of hospitals available in the region/state; identification of physical and personal resources available in outpatient care in the region/state) [57], France (regional strategic health projects in the field of prevention, treatment, and social assistance enriched by identifying the distribution of medical equipment; regional health plan for the most excluded) [58], and Spain (identification of various problems as a consequence of preparing separate autonomous studies for communities and cities) [59].

In the area of planning health needs, and then healing processes, it is important to rely on available equipment resources, which are extremely important in relation to oncological diseases. Among other things, thanks to advanced imaging diagnostics, one can correctly recognise cancer, determine its stage, choose the appropriate treatment method and monitor the effectiveness of therapy, and detect any possible recurrence early. The time when the diagnostic test is performed is particularly important. Inequalities in access to diagnostic equipment shown in the maps have become the basis for taking active actions in the field of health policy. Contrary to popular belief, these activities were not just about buying missing equipment but actually verifying the demand. These types of activities, inspired by information contained in the maps of health needs, allow decision-makers to make more informed decisions and thus to be more effective in the actual levelling of development differences.

Purchases of diagnostic equipment for oncological services in Poland in recent years have been mainly based on financial resources from the European Social Fund. Directions for such activities are part of the EU strategic documents. The directions of financial support to improve access to medical infrastructure include the Operational Program Knowledge, Education and Development (PO WER), which includes activities supporting cancer prophylaxis supplemented with the opportunity to purchase the missing equipment necessary to complete the task. About 60 million PLN was devoted for this purpose under PO WER 2014–2020, of which the allocation for the head and neck cancer prevention program was 15 million PLN; for the skin cancer program, 15 million PLN; lung cancer prevention, 20 million PLN; and the program in in hereditary cancers, 10 million PLN [60].

In addition, the medical infrastructure was supplemented on the basis of Voivodship Operational Programs for 2014–2020 inscribed in relevant priority axes, e.g., Priority Axis 6. Social cohesion infrastructure and specific actions in the field of 6.2 Investments in health infrastructure (Lower Silesian Voivodeship). The main determinant of activities in this area is the document of the National Strategy for Regional Development 2010–2020: “Regions, cities, rural areas”, i.e., “effective use of specific regional and other territorial development potentials for achieving the country’s development goals – growth, employment and cohesion in the long-term horizon.” Objective 2, “Building territorial cohesion and counteracting marginalisation of problem areas” perfectly fits action 2.2.2., “Improving the quality of access to medical services in problem areas.” Activities in this area are aimed at improving the health condition of their residents, which directly translates into the number of professionally active people and the quality and efficiency of their work.

Medical infrastructure also finds a source of power in the Operational Program Infrastructure and Environment 2014–2020, where Measure 9.2 Infrastructure of supra-regional medical entities enables the support of wards and other organisational units of supra-regional hospitals that are focused on the treatment of diseases affecting occupational deactivation (e.g., cancer). In total, 121 million PLN has been allocated to these areas.

Oncology is also supported from non-EU European funds through Norwegian and EEA funds [61]. These are the PL07 program, “Improving and better adapting healthcare to demographic and epidemiological trends”, including five projects covering: Research, campaigns to promote a healthy lifestyle, retrofitting and modernising medical entities; as well as the PL13 program, “Reducing social inequities in health”, including 26 projects, of which seven are on cancer prevention (co-financing of preventive programs, educational and promotional activities). All these activities are intended to strengthen the resources of Polish healthcare in order to improve diagnostics, treatment, and access to services, and, consequently, reduce the social costs of oncological diseases (in their broad sense).

## 4. Conclusions

The increase in the incidence of oncological diseases in the world over the past several years is indisputable. This phenomenon has resulted in the need for governments and entire societies to incur increasingly higher costs related to healthcare, including oncological care. An alternative to the lack of increasing involvement in the process of preventing the spread of cancer would be a very marked increase in mortality, increased disability, and loss of productivity. In view of the above, it is urgent to take action to stop negative tendencies. There are efforts being carried out in parallel in the medical field, where the most effective methods of therapy and drug programs are sought, and in the management field, for which the health policy of the state is of fundamental importance.

Cancer rates recorded in Poland put us at the forefront of the list. Calculating the direct costs of cancer (based on bills from the payer) is not a particularly difficult task, although very arduous. On the other hand, calculating the social cost (broadly understood) referring to even the most frequently occurring oncological diseases would not only be arduous but also very difficult due to the limited availability of data and the applied classification diversity of said data. However, single selected studies conducted for selected oncological diseases show the huge scale of the problem. These studies, together with the arguments of representatives of the medical community, have become the basis for healthcare policy makers to undertake a number of positive actions. However, the real effectiveness of the actions taken in Poland will be visible at the soonest in a few years. Diagnostic oncology card (DILO), maps of healthcare needs, and strengthening the medical infrastructure of healthcare entities will all bring positive effects, provided they are used in practice. Additionally, while the DILO card is currently used by oncologically ill patients, maps of healthcare needs are still awaiting their introduction to the current plans and activities.

The European Union, together with associated countries, acting as an entity shaping the foundations of socioeconomic development, also directs its interest in supporting activities for sustainable development and harmonious economic activity, including support for actions to level development differences.

In view of the above, it is clear that the problem of health and disease from a socioeconomic perspective must also be re-evaluated, the more so that economists have not been interested in the value of health for a long time, but have only studied how the health of society affects economic growth and how a healthy society can contribute to the economic development of individual countries [46]. Falling sick alone limits the production capacity of individual units (loss of productivity) while the demand for additional help increases (additional social cost in the broad sense). As a result, we are dealing with unplanned costs, the amount of which varies depending on the type of disease or its stage of development. The disease also affects the ability to work. Absence from work or reduced performance of a sick employee, in turn, reduces the quality and number of tasks performed (lost production). The sum of such effects observed at the level of all enterprises reflects the costs borne by the whole economy. Thus, all factors lowering the level of global health, and in particular the causes of a long-term reduction in the health of the population, such as chronic oncological diseases, impact health as one of the main pillars of the competitiveness of the economy [62].

## Figures and Tables

**Table 1 ijerph-17-02837-t001:** Economic cost, social cost: An overview of terms.

The Type of Cost	Definition/Interpretation; Cost Components	Sample Literature
Economic cost	Cost from the perspective of the enterprise; allows to determine the economic efficiency of operations; to determine in which sector of the economy it is best to allocate the resources. In economic activity, it is a type of production cost; Economic cost = accounting costs + opportunity costs + normal profit, or: Economic cost = accounting costs + opportunity costs where: (1) Accounting costs (explicit - visible) are production costs expressed by multiplying the number of factors purchased by their price expressed in money are recorded in the company’s accounting. (2) Alternative costs, i.e. lost costs (implicit - invisible) - allow you to determine the benefits that could be achieved by allocating resources to other activities (3) Normal profit - the minimum remuneration that an entrepreneur must receive in order to continue a given undertaking.	[10,11,12,13]
Social costs	All direct and indirect losses, i.e. all harmful consequences and damages suffered by third parties or the entire society as a result of the economic activities of individual producers and for which it is not easy to blame individual legal and physical entities.	[14,15,16,17,18,19]
Social cost (in general)	Social cost is the **economic** cost to society as a whole; a sum of private cost and external costs. Private cost is the economic cost incurred by private entities (enterprises and households), measurable and immeasurable; Social cost = private cost (measurable and immeasurable) + External costs	[20]
External costs = social costs	A perspective beyond the entrepreneur’s perspective: Harmful (negative) externalities of production processes. These are costs incurred by persons not directly involved in the production, consumption or exchange of a given good; they are therefore adverse effects of business activity experienced by third parties. It is in the social interest to reduce the negative external effect (by means of an adjustment fee, similar to the Pigou tax) to a level at which the level of external costs-social costs will be lower than social benefits. External effects can be negative (costs) and positive (benefits)	[21,22]
Social cost (in healthcare) - broad approach	**economic** cost to the whole of society; the sum of direct costs (medical and non-medical), indirect costs (related to reduced productivity) and non-measurable costs (hardly measurable in monetary terms, e.g. the cost of occupational non-fulfilment or the cost of pain) social cost (broad approach) = direct costs + indirect costs + non-measurable costs	[23,24,25]
The so-called social cost (in healthcare) – narrow approach	Immeasurable costs recognised as a category of other types of costs in healthcare. Other costs = immeasurable costs = so-called social cost	[26,27]
Relationship between social cost (broadly defined), social cost (in general) and economic cost	**Direct costs** as the sum of accounting costs incurred by healthcare providers, producers of medical and non-medical supplies used for healthcare purposes, associated service providers (e.g., transport, third party care) and private costs of non-economic natural persons (third party care) direct costs = accounting costs (of all entrepreneurs) + measurable private cost (of non-entrepreneurs) **Indirect costs** are the costs of lost opportunities to achieve greater social benefits (lost opportunities to increase productivity), the effect is lower productivity indirect costs = lost social benefits (lost positive externalities related to health productivity) The so-called social cost is the immeasurable cost of health incurred by human individuals, i.e., unmeasurable private costs (e.g., the cost of occupational failure or the cost of pain). The so-called social cost = immeasurable private costs (of all members of society) Social cost (broad approach) = direct costs + indirect costs + so-called social cost Social cost = (accounting costs + measurable private cost) + indirect costs + measurable private cost	Own interpretation
Social costs (as defined by WHO)	Social costs are a loss of social well-being resulting from lost production (indirect costs as defined by the WHO) as well as from costs that are difficult to measure in monetary units. Social costs = indirect costs (value of lost production) + non-measurable costs	[28]
Relationship between social costs (in the WHO sense), social costs (in the general sense), social cost (in the broad sense)	Social costs (WHO) = lost social benefits (i.e., lost production) + unmeasurable private costs Social costs (WHO) = lost social benefits (i.e., lost production) + so-called social cost Social costs (WHO) = social costs (in general) - direct costs	Own interpretation

Source: own.

**Table 2 ijerph-17-02837-t002:** Expenditure on healthcare and cancer care in selected countries.

Country	GDP per Capita EUR ^(a)^	GDP per Capita (EUR PPP ^(b^)	Healthcare Expenditure as % of GDP	Expenses for Healthcare per Capita EUR	Expenses for Healthcare per Capita (EUR PPP)	Expenses for Oncology as % of Health Expenses ^(c)^	Expenditure on Oncology per Capita (EUR)	Expenditure on Oncology per Capita (EUR PPP)
USA	40,000	38,800	17.7	7080	6868	4.7	333	323
Great Britain	29,800	26,800	9.4	2801	2519	6.1	171	154
Norway	75,700	49,700	9.4	7116	4672	2.5	178	117
France	31,300	27,700	11.6	3631	3213	4.3	156	138
Czech Republic	14,200	20,700	7.5	1065	1553	8.0	85	124
Poland	10,100	17,100	6.9	697	1180	6.0	42	70

Source: (a) Eurostat data for 2013 or 2012 (in the case of data including purchasing power), (b) OECD data for 2011 (2012 for France and Norway), (c) Data for Poland and the Czech Republic for 2011, for the USA and Great Britain for 2010 for Norway for 2007 and for France the average of values from various sources from 2009–2013. Based on: Cancer Research UK, Cancer Service: Reverse, Pause or Progress, December 2012, Institute for Fiscal Studies, Public payment and private provision, Nuffield Trust, May 2013, R. Luengo-Fernandez et al., Economic burden of cancer across the European Union: a population-based cost analysis, University of Oxford, October 2013, The National Cancer Institute, Cancer Trends Progress Report—2011/2012 Update, NIH, DHHS, Bethesda, MD, August 2012, http://progressreport.cancer.gov, SINTEF, Costs of cancer in the Nordic countries—a comparative study of health care costs and public income loss compensation payments related to cancer in the Nordic countries in 2007; Société Française de Radiothérapie Oncologique: Livre blanc de la radiothérapie en France, 2013; INCa (red.), Les cancers en France en 2013. Collection état des lieux et des connaissances, Boulogne-Billancourt Cedex, Styczeń 2014 oraz CNAMTS, Améliorer la qualité du système de santé et maîtriser les dépenses: propositions de l’Assurance Maladie Rapport au ministre chargé de la sécurité Sociale et au parlement sur l’évolution des charges et produits de l’assurance maladie au titre de 2014 (loi du 13 août 2004) pour 2014 oraz Economic information on health care, Zdravotnická Statistika ČR 2012, www.uzis.cz. For: Healthcare systems in selected countries, EY Report commissioned by the Oncology Foundation, 2014.

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
