# Peer review of "Oncological Diseases and Social Costs Considerations on Undertaken Health Policy Interventions"

_ijerph, 2020, doi:10.3390/ijerph17082837_

Round 1

Reviewer 1 Report

Is this cost effectiveness analysis study? I couldn't any cost list or effectiveness list at all. and I couldn't get any information from this study regarding cancer care cost neither. There is no explains regarding analysis the cost and effectiveness using data.

Author Response

Dear Reviewer, thank you very much for reading the article and for the offered review. The answers to the indicated comments and remarks regarding the article are presented below.

1. The article did not foresee any profitability analysis and did not formulate either a list of costs or a list of effectiveness. There are no calculations regarding the cost of cancer treatment, either.

The resulting comments are probably the result of an incorrectly formulated title of the paper (inadequate to the actual content) and the result of an over-generalisation of its goal.

The goal of the article was to identify actions undertaken by health politicians to reduce the cost of oncological diseases (in particular social costs) incurred in Poland, and to evaluate these actions. Empirical data used in the study were secondary data (they did not come from own research). The set goal was achieved by analysing the literature on the subject and statistical data, followed by induction based on it.

The new (corrected) title of the article is: “Oncological diseases and social costs. Considerations on undertaken health policy interventions.”

2. Thank you very much for the generous suggestion of the literature sources.

The suggested literature focuses essentially on three areas. The first area is epidemiology, including modelling mortality, estimating disease incidence, and five-year survival; the second area: calculating the cost of oncological diseases (mainly direct costs) of various organs; the third area: information on health spending levels in various regions of the world, including Eastern Europe. Nearly every suggested item of literature turned out to be valuable because it found its use in the reviewed article. The article has also been supplemented with additional sources.

We hope that the presented answer is sufficient and justifies the authors’ considerations and reasoning. We remain with all due respect, Authors.

Reviewer 2 Report

The subject of the article is extremely important given the individual and population health effects and economic consequences of the cancer epidemic. Therefore, I recommend the publication of the text.

However, the authors should make a few improvements.

In particular, they should consider changing the title of the article for two reasons.

Firstly, current title "Economic and social costs in oncological diseases on the example of Poland, including the support of the European Union" suggests that the article is devoted to the analysis of costs related to the occurrence of cancer in Poland (at macro level). However, the information on this subject takes only few lines of the text (lines 206-226). Significantly more attention (and rightly so) is paid by the authors to activities that have been undertaken in recent years and tools that are used to improve access to oncological services (both diagnostic and therapeutic) and to improve resource allocation.

Secondly, the current title suggests a duality of cost categories - a duality that is not explained in the text.

The above recommendation is connected with another one concerning more precise definition of economic and social cost categories. The confusion in the scientific literature in this respect does not facilitate this task (different authors adopt different conventions). Nevertheless, the authors of the reviewed text should adopt some consistent classification. Currently, the title lists two cost categories - economic and social. In the abstract - line 24 - only measurement of social costs of cancer exceeding the costs of treatment is mentioned. In line 49 the notion of indirect cost appears, which (as one can guess - has not been specified) and concerns the loss of GDP due to the occurrence of various diseases. However, the sentence "Premature deaths and the need for long-term care directly affect the value of these costs" (lines 53 and 54) is not precise. While premature deaths may be associated with productivity losses, the emergence of the need for long-term care is not necessarily anymore - unless it is provided, for example, by informal caregivers - family members resigning from labor market.

In the further part of the article, the notion of indirect cost appears more often (e.g. from line 85) as a supplement to the costs of using specific health technologies (“The analysis of the cost calculation process should not be limited to taking into account the costs of using specific health technologies, i.e. the amount of expenditure related to treatment and possibly prevention”). Economic and social costs are mentioned again. Economic costs seem to refer to the superior category - "These costs are classified as: direct costs (medical and non-medical: e.g. transport, third party care), indirect (related to reduced productivity) and non-measurable, i.e. difficult to measure in monetary terms (e.g. the cost of occupational failure or the cost of pain)” (lines 94-96). In turn, social costs are identified with non-measurable costs - "Non-measurable costs are often recognized in the category of other types of costs in healthcare and are referred to as so-called social costs.” (lines 97-98). In the next sentence, referring to WHO, social costs are defined as the sum of non-measurable costs and indirect costs (lines 99-102). Starting from line 106, the authors mainly deal with the ways of measuring indirect costs, which, however, cannot be identified with social costs in the sense of WHO definition, nor in a narrower sense as non-measurable costs. Social costs appear again only in line 204, whereas the notion of economic costs does not appear in principle since line 94.

I also recommend a careful review of the paper in terms of using economic categories.

E.g. in the sentence "By supplementing these values with the costs of preventive programs, estimated indirect costs, including the costs of lost production in the economy resulting from the occurrence of cancer, and the costs associated with broadly understood social security, e.g. costs of sick leave, we can see that the level of total expenditure is many times higher (lines 214-218). A mistake in this sentence refers to the fact that all costs are identified with expenditures, but costs of lost production in the economic perspective are not expenditures.

Also the sentence ”Therefore, expenditure on treatment cannot be treated as a cost, but as a broadly understood investment not only in health, but also in shaping the foundations of development and economic growth”' (lines 42-43) is questionable, although it is now commonly expressed in Polish public debate. It artificially contrasts the incomparable categories of costs and investments - as if the cost were by definition something inappropriate, wrong, generating losses and the investment something good, worthy of support. In fact, the production of any investment good requires using of resources and incurring costs.

The authors should also supplement references - especially to lines 97-99, 125-135 and 136-144 (description of methods), 210-216, 285-287.

Author Response

Dear Reviewer, thank you very much for reading the article as well as for the prepared in-depth review and constructive comments.The answers to the indicated comments and remarks regarding the article are presented below.

1. The new (corrected) title of the article is: “Oncological diseases and social costs. Considerations on undertaken health policy interventions.”

2. In order to clarify individual concepts and definitions, a table has been prepared showing their different approaches relevant to the considerations undertaken in the article. Therefore, redundant definition elements contained in the article have been removed. The details are presented in the modified table contained in the pdf document - under the name Reviewer 2 and in the main text.

We hope that the provided answers are sufficient, and the changes introduced to the text significantly increase the quality of the considerations contained in the article. We remain with all due respect, Authors.

Round 2

Reviewer 1 Report

Thank you for your revision according to my suggestion.

I felt the manuscript got more fluency.